

# Molecular phylogeny and evolutionary history of *Moricandia* DC (Brassicaceae)

Francisco Perfectti[1], José M. Gómez[2,3], Adela González-Megías[4], Mohamed Abdelaziz[1] and Juan Lorite[5]

[1] Departamento de Genética, Universidad de Granada, Granada, Spain
[2] Departamento de Ecología Funcional y Evolutiva, Estación Experimental de Zonas Áridas, CSIC, La Cañada de San Urbano, Almería, Spain
[3] Departamento de Ecología, Universidad de Granada, Granada, Spain
[4] Departamento de Zoología, Universidad de Granada, Granada, Spain
[5] Departamento de Botánica, Universidad de Granada, Granada, Spain

Corresponding author
Francisco Perfectti, fperfect@ugr.es

## ABSTRACT

**Background.** The phylogeny of tribe Brassiceae (Brassicaceae) has not yet been resolved because of its complex evolutionary history. This tribe comprises economically relevant species, including the genus *Moricandia* DC. This genus is currently distributed in North Africa, Middle East, Central Asia and Southern Europe, where it is associated with arid and semi-arid environments. Although some species of *Moricandia* have been used in several phylogenetic studies, the phylogeny of this genus is not well established.

**Methods.** Here we present a phylogenetic analysis of the genus *Moricandia* using a nuclear (the internal transcribed spacers of the ribosomal DNA) and two plastidial regions (parts of the NADH dehydrogenase subunit F gene and the *trn*T-*trn*F region). We also included in the analyses members of their sister genus *Rytidocarpus* and from the close genus *Eruca*.

**Results.** The phylogenetic analyses showed a clear and robust phylogeny of the genus *Moricandia*. The Bayesian inference tree was concordant with the maximum likelihood and timing trees, with the plastidial and nuclear trees showing only minor discrepancies. The genus *Moricandia* appears to be formed by two main lineages: the Iberian clade including three species, and the African clade including the four species inhabiting the Southern Mediterranean regions plus *M. arvensis*.

**Discussion.** We dated the main evolutionary events of this genus, showing that the origin of the Iberian clade probably occurred after a range expansion during the Messinian period, between 7.25 and 5.33 Ma. In that period, an extensive African-Iberian floral and faunal interchange occurred due to the existence of land bridges between Africa and Europa in what is, at present-days, the Strait of Gibraltar. We have demonstrated that a Spanish population previously ascribed to *Rytidocarpus moricandioides* is indeed a *Moricandia* species, and we propose to name it as *M. rytidocarpoides* sp. nov. In addition, in all the phylogenetic analyses, *M. foleyi* appeared outside the *Moricandia* lineage but within the genus *Eruca*. Therefore, *M. foleyi* should be excluded from the genus *Moricandia* and be ascribed, at least provisionally, to the genus *Eruca*.

## INTRODUCTION

The Brassiceae tribe of the Brassicaceae family includes many economically important species that are useful as vegetables, edible oils, crop forages, condiments and fuel crops (*Zelmer & McVetty, 2009*). For this reason, this crucifer tribe has been the focus of a vast amount of genetic, agronomic, and ecological research (*Gómez-Campo, 1999*; *Gupta, 2009*; *Kole, 2011*; *Schmidt & Bancroft, 2011*). Despite this interest, the phylogenetic relationship between the Brassiceae species is not yet resolved, and consequently, several well-established Brassiceae genera, such as *Brassica* or *Diplotaxis*, are probably polyphyletic and artificial. One small but economically important genus in this tribe is *Moricandia* DC. This genus probably originated in the Mediterranean Basin (*Pratap & Gupta, 2009*) and is currently distributed in North Africa, Middle East, Central Asia and Southern Europe, where it is associated mostly with arid and semi-arid environments (*Tahir & Watts, 2011*). Some *Moricandia* species have being extensively studied because they show intermediate $C_3$–$C_4$ photosynthetic metabolism (*Apel, Horstmann & Pfeffer, 1997*; *McVetty, Austin & Morgan, 1989*), a feature that improves carbon assimilation and water use efficiency under drought conditions (*Apel, Bauve & Ohle, 1984*; *McVetty, Austin & Morgan, 1989*). This so-called *Moricandia syndrome* may have great agronomic interest since it could be transferred to *Brassica* species by hybridization, increasing crop yield under extreme climatic conditions and in marginal areas (*Apel, Bauve & Ohle, 1984*). Disentangling the phylogenetic relationship between *Moricandia* species is key to understanding how these traits have evolved and to determine the placement of this genus inside the Brassiceae tribe.

*Moricandia* individuals mainly show erect and branched stems with simple, exstipulate leaves, usually with entire or pinnated lobes (*Gupta, 2009*). Their flowers are actinomorphic-disymmetric and mostly of lilac color, although range from almost white to deep purple depending on the species and weather conditions. Their fruits are dehiscent two-valves siliques with one or two seed series per valve (*Gupta, 2009*). *Moricandia* shows a high variability in the morphological characters used for identification, making the taxonomy of this genus complex and controversial (*Jiménez & Sánchez Gómez, 2012*). Eight species are currently recognised in the genus *Moricandia* (*Warwick & Sauder, 2005*; *Warwick, Francis & Al-Shehbaz, 2006*; *Tahir & Watts, 2011*): (1) *M. arvensis* (L.) DC, (2) *M. moricandioides* (Boiss.) Heywood, (3) *M. foetida* Bourg. ex Coss., (4) *M. suffruticosa* (Desf.) Coss. & Durieu, (5) *M. spinosa* Pomel, (6) *M. foleyi* Batt., (7) *M. sinaica* Boiss., and (8) *M. nitens* (Viv.) Durieu & Barr (*Sobrino Vesperinas, 1984*; *Warwick, Francis & Al-Shehbaz, 2006*; *Tahir & Watts, 2011*). *Moricandia arvensis* is an annual to perennial herb widely distributed in the northwest Africa, Iberian Peninsula and southern Italy, from where it has even invaded other parts of the planet (*De Bolós, 1946*; *Gómez-Campo, 1999*). It is mostly a ruderal species associated to cultivated areas, roadsides and other human disturbed habitats. *Moricandia moricandioides* and *M. foetida* are two herbaceous species endemic to the Iberian Peninsula. The former is distributed in semi-arid environments of the eastern Spain whereas the latter is a narrow endemism inhabiting arid habitats of the southeast Spain (*Sobrino Vesperinas, 1993*). *Moricandia suffruticosa and M. spinosa* are suffruticose species inhabiting Morocco, Algeria, Tunisia and probably Libya, whereas

*M. foleyi* is an annual herb showing a very narrow distribution in desert areas of southern Morocco and Algeria. *Moricandia sinaica* is a herbaceous species located in desert areas from the Near East to Pakistan; and *M. nitens* is a suffruticose species distributed from North Africa to Middle East. In addition, several subspecies and varieties have been named, in particular for the widely distributed *M. arvensis* (*Schulz, 1936*; *Heywood, 1964*; *Maire, 1967*).

The phylogenetic relationships among the *Moricandia* species, and of these with the rest of the Brassiceae species, have not been well established yet. *Moricandia* was previously included into the subtribe Moricandiinae (*Schulz, 1923*; *Schulz, 1936*; *Gómez-Campo, 1980*), although molecular evidences have shown little support for this subtribe (*Warwick & Black, 1994*; *Warwick & Sauder, 2005*). Most family-wide phylogenies have included some *Moricandia* species (e.g., *Warwick & Black, 1997*; *Warwick & Sauder, 2005*; *Bailey et al., 2006*; *Jiménez & Sánchez Gómez, 2012*; *Schlüter et al., 2016*), and these large-scale studies have suggested that *Moricandia* belongs to the Rapa/Oleracea subtribe together with *Brassica*, *Diplotaxis*, *Enartrocarpus*, *Eruca*, *Erucastrum*, *Morisia*, *Raphanus*, *Rapistrum* and *Rytidocarpus* (*Warwick & Hall, 2009*). Although *Moricandia* has been considered a *Brassica* coenospecies on the basis of cytogenetic and morphological similarities (*Gómez-Campo, 1999*), molecular phylogenetic analyses suggest that *Moricandia* is closely related to the genus *Rytidocarpus* and less clearly to *Eruca* (*Warwick & Black, 1994*; *Warwick & Sauder, 2005*; *Bailey et al., 2006*; *Couvreur et al., 2010*). The monotypic genus *Rytidocarpus* has long been recognized as very close to *Moricandia* because it presents *Moricandia*-like cotyledons, similar purple flowers and the same chromosome ($2n = 28$) complement (*Gómez-Campo, 1980*; *Prakash et al., 2009*). Thereby, due to this incomplete and sometimes contradictory evidence, the phylogenetic position and evolutionary history of *Moricandia* is still unresolved, despite the importance this information could have to understand the evolution of agronomic traits.

The main goal of this study is to disentangle the phylogenetic relationships among the *Moricandia* species, using a nuclear and two plastidial regions. In addition, we have dated the main events in the evolution of this genus and determined the phylogenetic relationship of *Moricandia* with its closely related genera *Rytidocarpus* and *Eruca*. We show that one *Moricandia* species should be excluded of this genus and demonstrate that a population previously ascribed to *Rytidocarpus moricandioides* is indeed a new *Moricandia* species.

## MATERIALS AND METHODS

### Taxon sampling

We collected leaf tissue of 1–5 individuals from a total of 17 populations of *Moricandia* (Table 1), including five populations of *Moricandia arvensis*, five populations of *M. moricandioides* (two of the subspecies *pseudofoetida* and one of each subspecies *baetica*, *giennensis*, and *moricandioides*), two populations of *M. foetida*, two of *M. foleyi*, and one of each *M. nitens*, *M. suffruticosa* and *M. spinosa*. In addition, we included two populations of *Eruca pinnatifida* and one of *E. vesicaria*. Four populations of *Rytidocarpus moricandioides*, two from Morocco and one from Spain (sampled two consecutive years), were also included

**Table 1** Taxa included in the 24-samples set with indication of population code, location of sampling, and reference herbarium voucher.

| Taxon | Population code | Voucher | Location | Geographical coordinates |
|---|---|---|---|---|
| *Moricandia arvensis* | Mar01 | GDA62592 | Barranco del Espartal, Baza, Granada, Spain | 37°31′12″N 2°42′11.99″W |
| | Mar33 | GDA62641 | Road between Santa Fe-La Malahá, Granada, Spain | 37°8′24″N 3°43′53.99″W |
| | Mar35 | MA321698-1 | Cortijo de las Monjas, Olula del Rio, Almería, Spain | 37°22′18″N 2°17′53.99″W |
| | Mar38 | MA50245-1 | Road to Mejorada del Campo, Madrid, Spain | 40°22′57.47″N 3°35′28.61″W |
| | Mar42 | MA321231-1 | Close to Road A-35, Mogente, Valencia, Spain | 38°52′17.22″N 0°46′23.52″W |
| *M. foetida* | Mfo01 | GDA49837-1 | Road between Tabernas and Sorbas, Almería, Spain | 37°0′15.83″N 2°27′26.76″W |
| | Mfo02 | GDA62639 | Olula del Rio, Almería, Spain | 37°22′18″N 2°17′53.99″W |
| *M. foleyi* | Mfy01[a] | GDA62595 | Road between Rissani and Merzuga, Morocco | 31°16′53.99″N 4°16′30″W |
| | Mfy02[a] | GDA62593 | Merzuga, Morocco | 31°3′30″N 4°0′42″W |
| *M. moricandioides baetica* | Mmob01 | GDA 62596 | Barranco del Espartal, Baza, Granada, Spain | 37°31′12″N 2°42′11.99″W |
| *M. moricandioides giennensis* | Mmog06 | GDA62638 | Road between Quesada and Huesa, Jaén, Spain | 37°45′57.18″N 3°12′8.09″W |
| *M. moricandioides moricandioides* | Mmom05 | GDA62640 | Road between Santa Fe-La Malahá, Granada, Spain | 37°8′24″N 3°43′53.99″W |
| *M. moricandioides pseudofoetida* | Mmsf01 | MUB105856 | Near Puerto del Garruchal, Murcia, Spain | 38°6′59.99″N 1°21′0″W |
| *M. moricandioides pseudofoetida* | Mmsf02 | – | Pago del Barranco y de Chumilla, Murcia, Spain | 37°56′14″N 1°1′58″W |
| *M. nitens* | Mni03 | GDA62597 | Close to Agouim, Morocco | 31°10′7.2″N 7°29′15.72″W |
| *M. spinosa* | Mspi01 | GDA62598 | Road between Missour-Boulemane, Morocco | 33°2′8.63″N 4°4′4.98″W |
| *M. suffruticosa* | Msu01 | GDA62599 | Road between Taza and Aknour, Morocco | 3°23′50.16″N 3°54′24.77″W |
| *Eruca pinnatifida* (=*E. vesicaria pinnatifida*) | Erupinn01 | GDA62602 | Road between Missour and Boulemane, Morocco | 33°2′8.63″N 4°4′4.98″W |
| | Erupinn02 | – | Merzuga, Morocco | 31°3′30″N 4°0′42″W |
| *E. vesicaria* (=*E. vesicaria vesicaria*) | Eruves01 | GDA62643 | Barranco del Espartal, Baza, Granada, Spain | 37°31′12″N 2°42′11.99″W |

**Table 1** (*continued*)

| Taxon | Population code | Voucher | Location | Geographical coordinates |
|---|---|---|---|---|
| *Rytidocarpus moricandiodes* | Rmorm01 | GDA62600 | Road Taza-Aknour, Morocco | 3°23′50.16″N 3°54′24.77″W |
| | Rmorg02 | GDA62601 | Moulay Yacoub, Morocco | 34°7′27.12″N 5°12′14.87″W |
| | Rmorg01-13[b] | GDA62636 | Road A322, close to Quesada, Jaén, Spain | 37°50′36.73″N 3°8′22.17″W |
| | Rmorg01-14[b] | GDA62636 | Road A322, close to Quesada, Jaén, Spain | 37°50′36.73″N 3°8′22.17″W |

**Notes.**
[a]Proposed as *Eruca foleyi*.
[b]Proposed as *Moricandia rytidocarpoides*.

as representatives of this monotypic genus, which is probably the sister genus of *Moricandia* (*Warwick & Sauder, 2005*). These samples constituted the 24-samples set. Table 1 shows the code and location of these populations as well as the reference voucher material.

We also downloaded all the *Moricandia*-related ITS sequences hosted in GenBank (downloaded on March 1st, 2016). After quality-checking, we discarded those sequences that did not show complete ITS1 and ITS2 sequences and finally kept 18 *Moricandia*, 4 *Eruca* and 2 *Rytidocarpus* sequences for the following analyses. Specifically, we included seven *M. arvensis* (AY722472, DQ249832, EF601897, EF601898, EF601899, EF601900—var. *robusta*-, EF601901—var. *garamantum*-), one *M. foetida* (EF601902), one *M. foleyi* (EF601903), three *M. moricandioides* (AY722473, KF849875, EF601904), two *M. nitens* (AY722474, EF601905), one *M. sinaica* (EF601906), one *M. spinosa* (EF601907), two *M. suffruticosa* (AY722475, EF601908), two *R. moricandioides* (AY722483, EF601910), two *E. vesicaria sativa* (AY254536, DQ249821), one *E. vesicaria vesicaria* (AY722459), and one *E. pinnatifida* (AY722458) accessions.

## DNA extraction, PCR amplification, and sequencing

Leaf tissues were freshly sampled from the specimens and subsequently desiccated and preserved in silica gel until DNA extraction. For each individual sample at least 60 mg of plant material was disrupted with a Mixer Mill MM400 (Retsch, Haan, Germany) using 2 mm steel beads. DNA was extracted using the GenElute Plant Genomic DNA Miniprep Kit (Sigma-Aldrich, St. Louis, MO, USA) following manufacturer's instructions.

One nuclear and two chloroplast DNA regions were amplified and sequenced. The nuclear sequence was composed by the internal transcribed spacers of the ribosomal DNA (ITS1 and ITS2) and the *5.8 rDNA* between both ITSs sequences, together with partial 18S and 28S sequences, with jointly spans ∼700 base pairs. The plastidial regions span 2004 base pairs for the NADH dehydrogenase subunit F (*ndhF*) gene and ∼1,600 base pairs for the *trn*T-*trn*F region, including intergenic spacers.

ITS regions were amplified with primers ITS1, ITS2, ITS3 and ITS4 (*White et al., 1990*) anchoring to ribosomal flanking regions using the following PCR conditions: 3 min at 94 °C as initial denaturing step, followed by 35 cycles of 15 s at 94 °C, 30 s at 64 °C (ITS1–ITS2 primers) or 53 °C (ITS3–ITS4 primers) and 45 s at 72 °C, and a final step of 3 min at

72 °C. The *ndhF* region was amplified using primers ndhF5, ndhF599, ndhF1354, and ndhF2100 (*Taberlet et al., 1991*) and the following PCR conditions: 3 min at 94 °C as initial denaturing step, followed by 35 cycles of 15 s at 94 °C, 30 s at 47 °C (for both primer pairs: ndhF5–ndhF1354 and ndhF599–ndhF2100) and 90 s at 72 °C, and a final step of 3 min at 72 °C. The *trn*T-*trn*F region was amplified with primers tabA, tabD, tabC and tabF (*Taberlet et al., 1991*). The PCR conditions for these regions were 3 min at 94 °C as initial denaturing step, followed by 35 cycles of 15 s at 94 °C, 30 s at 53 °C (tabA–tabD primers) or 58 °C (tabC–tabF primers) and 90 s at 72 °C, and a final step of 3 min at 72 °C. All PCR reactions were performed in an Eppendorf™ S Mastercycler (Eppendorf, Hamburg, Germany). Amplicons were precipitated by centrifugation at 4 °C after the addition of 0.15 volumes of 3 M sodium acetate, pH 4.6, and 3 volumes of 95% (v/v) ethanol. Amplicons were sent to Macrogen Europe (Geumchun-gu, Seoul, Korea) for sequencing in both directions, using their corresponding PCR primers. Chromatograms were reviewed and contigs were produced using Geneious v. 9 (*Kearse et al., 2012*; Biomatters, Inc., San Francisco, CA, USA, http://www.geneious.com) and thorough revised and corrected by eye inspection. Sequences were uploaded to GenBank (accession numbers in Table S1).

## Sequence alignment and phylogenetic analysis

For the phylogenetic analyses we built two sequence sets: (1) the 24-samples set including the ITS, *ndhF* and *trn*T-*trn*F sequences for the sampled specimens, and (2) the GenBank-ITS set including the ITS sequences of the 24-samples set plus the 24 complete ITS sequences obtained from GenBank. We used *Brassica rapa* and *Raphanus sativus* sequences obtained from GenBank as roots. These genera belong to the same Rapa/Oleraceae lineage inside the tribe Brassiceae as *Eruca*, *Rytidocarpus* and *Moricandia* (*Warwick & Hall, 2009*). We used accessions KM538956 (*B. rapa*) and AY746462 (*R. sativus*) for ITS, and extracted the plastidial *ndh*F and *trn*T-*trn*F regions from the complete cpDNA accessions of both species (DQ231548 for *B. rapa* and K5716483 for *R. sativus*).

Sequences were aligned and concatenated with Geneious v.9 (Biomatters Ltd.) using the matff algorithm (*Katoh et al., 2002*) with posterior slight manual adjustments. Ribosomic 5.8S, and 18S and 28S flanking sequences did not show variation and were not included in the following procedures. Evolutionary substitution models were separately fitted for each DNA region using jModeltest 2.1.7 (*Darriba et al., 2012*). The best-fitted molecular evolutionary models under Bayesian Information Criterion were TPM3+G for ITS1, TIM2ef+G for ITS2, TPM1uf+I for *ndh*F, and TPM1uf+G for the *trn*T-*trn*F.

We reconstructed the phylogeny under Bayesian inference using MrBayes 3.2.1 (*Ronquist et al., 2012*). Two independent runs of six MCMC chains were run for $2 \times 10^6$ generations, sampling trees every 100 generations. Evolutionary models were implemented as GTR+G for ITS1, ITS2 and *trn*T-*trn*F, and as GTR with a proportion of invariable sites for *ndh*F. We checked convergence using Tracer v1.6.1 (*Rambaut et al., 2014*) and discarded, as the burn-in phase, the first 20% of the saved trees. The consensus tree was obtained with the remained trees. This process was performed with the 24-samples set for the entire

sequence concatenation, for only ITS sequences and for only cpDNA sequences (*ndh*F and *trn*T-*trn*F). In addition, we run MrBayes with the GenBank-ITS-set using the same parameter setting.

To compare the obtained Bayesian inference tree with other possible tree topologies, we performed Bayes factor analysis (*Kass & Raftery, 1995*), using MrBayes 3.2.1 to calculate the marginal likelihoods (estimated using stepping-stone sampling based on 50 steps with 39,000 generations (78 samples) within each step).

We used the RAxML software (*Stamatakis, 2014*) for maximum likelihood phylogenetic inference of the 24-samples set, using a partition model (ITS1, ITS2, *ndh*F and *trn*T-*trn*F) assuming GTR+G substitution models. We executed 100 initial fast parsimony inferences and thereafter a thorough ML search. For confidence analysis, a bootstrap of 100 replicates was also performed.

We checked whether the evolution of *Moricandia* was congruent with a molecular clock by using a likelihood ratio test as implemented in MEGA v.7 (*Kumar, Stecher & Tamura, 2015*). This test compares the ML values of the Maximum Likelihood tree obtained with and without assuming a molecular clock under a GTR evolutionary model.

To produce a dated chronogram we used Beast 2.0 (*Bouckaert et al., 2014*) following two approaches. First, we used the substitution rate for non-codifying plastidial DNA obtained from the literature (1.2–1.7 $\times$ 10$^9$ substitution/site/year; *Graur & Li, 2000*). Second, since the Iberian species appeared as monophyletic (see results), we used the Messinian Salinity Crisis (5.9 Ma) and the Zanclean Flood (5.33 Ma) as a calibration uniform interval for the separation of the Iberian species from the rest of the *Moricandia* genus. To obtain a more inclusive tree with all the *Moricandia* species, we included the *M. sinaica* ITS sequence obtained from GenBank in this approach. The Bayesian search for tree topologies and node ages were conducted during 20,000,000 generations in BEAST using the previously fitted substitution models and using a strict clock and a Yule process as priors. MCMC were sampled every 1,000 generations and used a burn-in of 10%. Appropriate sampling in the stationary phase was checked using Tracer v1.6.1 (*Rambaut et al., 2014*).

The electronic version of this article in Portable Document Format (PDF) will represent a published work according to the International Code of Nomenclature for algae, fungi, and plants (ICN), and hence the new names contained in the electronic version are effectively published under that Code from the electronic edition alone. In addition, new names contained in this work which have been issued with identifiers by IPNI will eventually be made available to the Global Names Index. The IPNI LSIDs can be resolved and the associated information viewed through any standard web browser by appending the LSID contained in this publication to the prefix "http://ipni.org/". The online version of this work is archived and available from the following digital repositories: PeerJ, PubMed Central, and CLOCKSS.

## RESULTS

Ribosomic 5.8S, and 18S and 28S flanking sequences did not show variation and were not included in the final alignment of the 24-samples set and the two outgroups, that

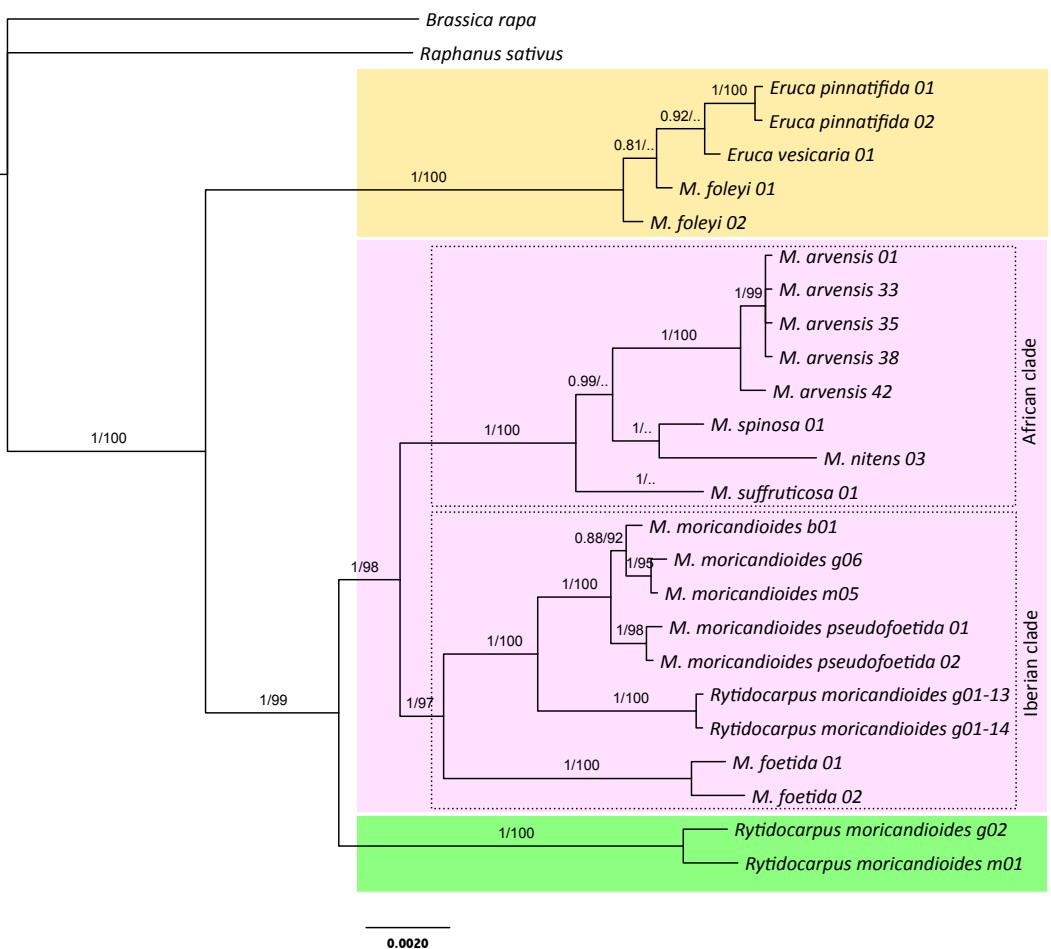

**Figure 1** **Bayesian inference tree produced with the complete 24-samples set.** In different color boxes are shown the genus *Eruca* (yellow), *Moricandia* (purple), and *Rytidocarpus* (green). The Iberian and African *Moricandia* clades are shown in doted line boxes. Branch labels are posterior probability values and bootstrap values obtained from the maximum likelihood phylogeny (Fig. S1). Scale in estimated substitutions per site. Note *Rytidocarpus moricandioides* samples g01-13 and g01-14 are proposed as the new species *Moricandia rytidocarpoides*.

spanned 4,055 bp. ITS1 extended over 274 bp and included 63 variable sites with 51 parsimonious informative positions. ITS2 spanned 194 bp and included 29 variable sites with 22 parsimonious informative positions. The amplified *ndh*F region spanned 2,004 bp with 87 variable sites and 58 parsimonious informative positions. No indels or stop codons were found on this region, which codifies for part of the NADH dehydrogenase subunit F. The *trn*T-*trn*F amplified region spanned 1,583 bp with 264 variable sites and 186 parsimonious informative positions.

## Join nuclear-plastidial phylogenetic trees

The Bayesian inference phylogeny for the complete 24-samples set, based on the nuclear and the plastidial regions, is shown in Fig. 1. The average standard deviation of split frequencies was 0.002 for the last generation, all estimated sample sizes were high (ESS > 960), and the
potential scale reduction factor (PSRF) was 1 for all the parameters, indicating good MCMC mixing and sampling. Convergence metrics clearly indicated that MCMC converged to a well-supported topology. In fact, posterior probabilities showed high values for branches separating the different species. In this tree, the genus *Moricandia* did not appear as a monophyletic clade. The two samples of the endemic *M. foleyi* were arranged into a clade also containing the samples from *Eruca* and showing a posterior probability of 1. We performed a Bayes factor analysis to compare this topology (H1: *Eruca* and *M. foleyi* forming a monophyletic clade) to those constraining *Eruca* and *M. foleyi* to different clades (H2). The marginal likelihoods (in natural log units) estimated with MrBayes using stepping-stone sampling were $-7987.71$ for H1 and $-8113.81$ for H2. This difference implies that H1 is strongly supported by the Bayes factor test. Therefore, the taxonomic status of *M. foleyi* should be reconsidered.

The rest of the *Moricandia* species formed two clades. The first one includes the five *M. arvensis* samples, together with *M. spinosa*, *M. nitens* and, more basal, *M. suffruticosa*. We called this clade as the 'African clade' because all these species inhabit in Africa. The second clade—the 'Iberian clade'—included the Spanish endemic species *M. moricandioides* and *M. foetida*, and the *Rytidocarpus* samples from Spain. The two samples belonging to subspecies *M. moricandioides pseudofoetida* appeared inside the branch of *M. moricandioides* but separated from the other *M. moricandioides* samples.

The genus Rytidocarpus appeared as the sister genus of *Moricandia*. The *R. moricandioides* samples from Morocco (Rmorg02-14-01 from Moulay Yacoub, close to Fez, and Rmorm01-14-01 collected in the road between the cities of Taza and Aknour; Table S1) form a monophyletic group. However, the *R. moricandioides* collected in Jaén, Spain, appeared inside the *Moricandia* Iberian clade, as a sister species of *M. moricandioides* (Fig. 1). To test the confidence of this result, we performed a Bayes factor analysis to test the topological hypothesis that the Spanish samples of *Rytidocarpus* are more closely related to *Moricandia* than to the other *Rytidocarpus* (Moroccan samples). Specifically, we compared the hypothesis that *Rytidocarpus* form a monophyletic group (H1; i.e., these Spanish samples are bona-fide *Rytidocarpus*) with the hypothesis that the two origins (Morocco and Spain) represent two different clades (H2). The marginal likelihoods (in natural log units) estimated with MrBayes using stepping-stone sampling were $-8037.09$ for H1 and $-7997.49$ for H2. This difference is very strong (decisive in the sense of *Kass & Raftery, 1995*) in favor of H2, and implies that Spanish sample should not be ascribed to *Rytidocarpus* but be recognized as a new species: *M. rytidocarpoides* (see Appendix for a formal description).

The ML tree produced with the 24-samples set (Fig. S1) was congruent with the Bayesian tree. They differed in the position of *M. suffruticosa* that appeared as the most basal species of the African clade, but with a low branch support. In addition, the position of *M. spinosa* and *M. nitens* also slightly varied with respect to the Bayesian tree, but these branching events were not supported with high bootstrap values (Fig. S1).
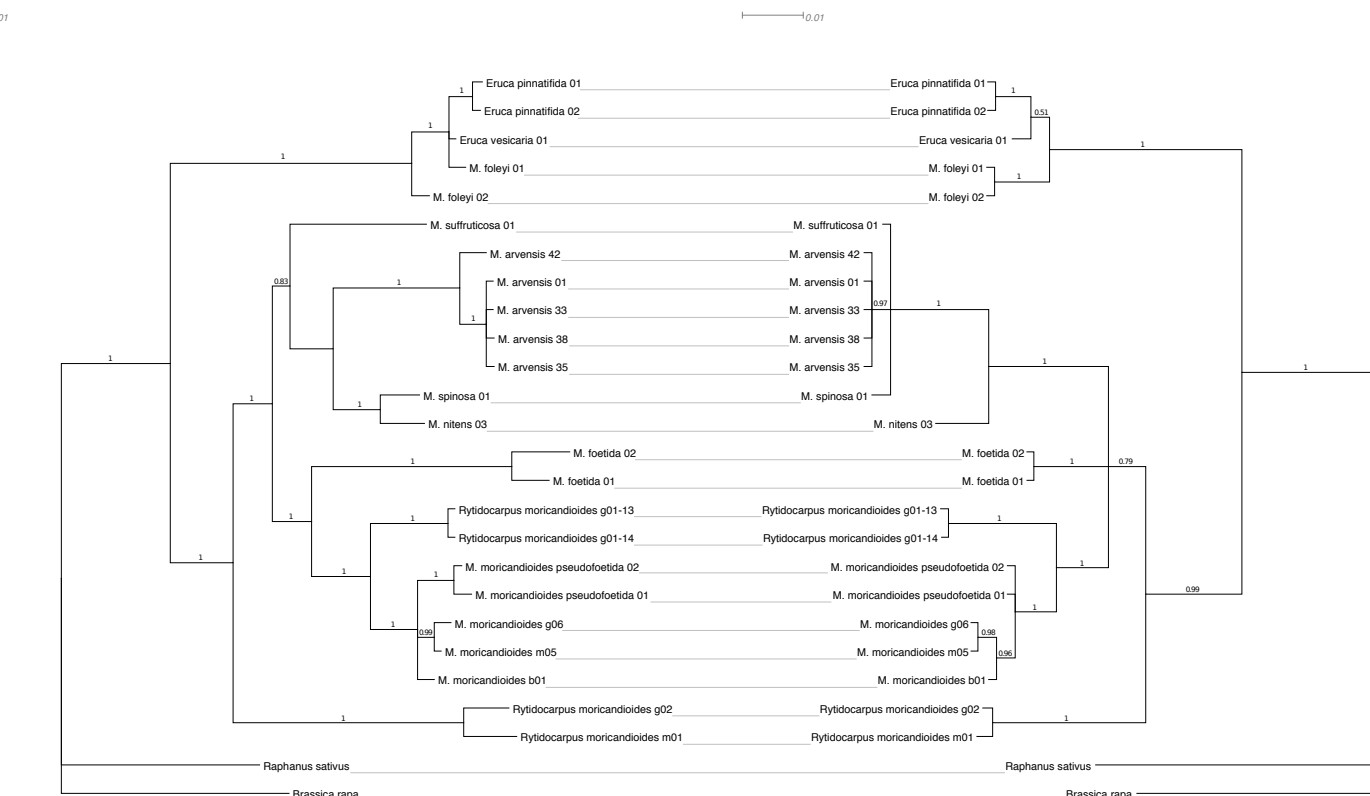

**Figure 2** **Tanglegram showing the Bayesian inference trees obtained from cpDNA (*ndhF + trnT-trnF*) and nuclear (*ITS1 + ITS2*) sequences.** Branch labels refer to posterior probabilities. Scale in estimated substitutions per site.

## Plastidial and nuclear phylogenetic trees

Figure 2 shows the Bayesian inference tree obtained with the combined cpDNA regions (*ndh*F and *trn*T-*trn*F) confronted with the tree obtained with the nuclear ITS regions using the same inference approach. The general pattern was maintained except for a lower resolution in the nuclear tree, where the *Moricandia* lineage appeared not so well resolved showing a basal trichotomy, and for minor rearrangements. For instance, *M. spinosa* and *M. nitens* appeared as sibling species in the plastidial tree but they were less closely related in the nuclear tree, although in both cases they belonged to the same African clade. It is also noticeable that *M. foleyi* samples appeared as monophyletic in the nuclear tree but interspersed with the *Eruca* samples in the plastidial tree. These trees were also obtained by ML inference and showed the same topologies (Fig. S2).

## ITS inclusive phylogeny

As a cross-validation of our sampling and to infer the phylogenetic position of *M. sinaica*, which is the only species of this genus that we could not sample, we used all complete ITS sequences available in GenBank to produce a comprehensive ITS phylogenetic tree (Fig. 3). The general pattern was compatible with our previous analyses. All our *Moricandia* samples were located close to GenBank ITS sequences of the same species. However, the

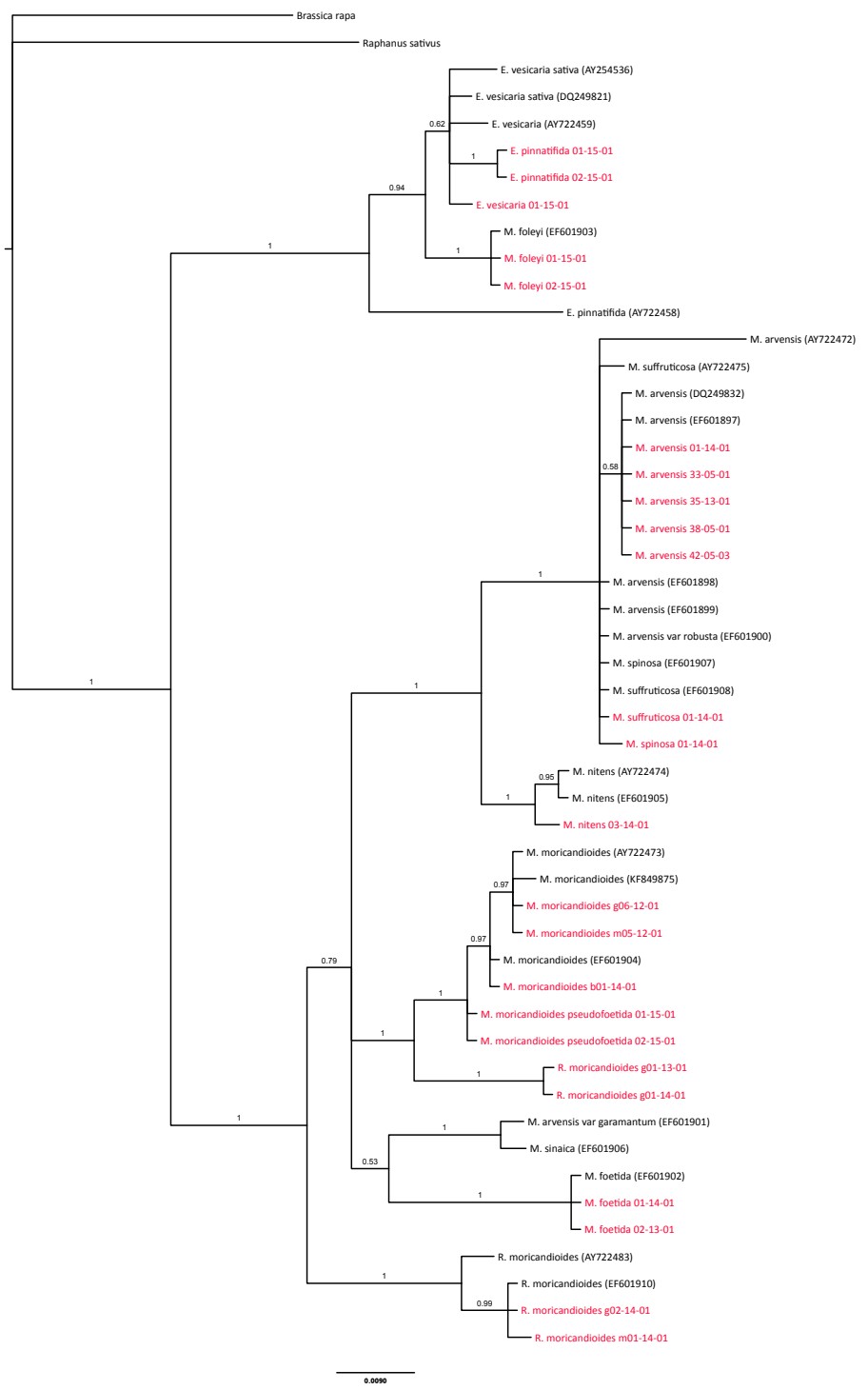

**Figure 3** **Phylogenetic tree based on ITS sequences from the ITS-set.** OTUs also included in the 24-samples set are depicted in red. Branch labels refer to posterior probabilities. Genbank accession codes are depicted between parentheses. Scale in estimated substitutions per site.

ITS tree showed lower resolution than the combined nuclear-plastidial tree. In the ITS tree, *Moricandia* showed a basal polytomy formed by four lineages. The first one included the *M. arvensis*, *M. suffruticosa* and *M. spinosa* accessions. The second lineage included the *M. moricandioides* accessions and the *R. moricandioides* sampled in Jaén, Spain. The third and fourth lineages appeared in a clade without enough support (posterior probability = 0.53) to be considered as a unique lineage. Therefore, the third lineage was formed by *M. sinaica* and *M. arvensis var. garamantum*, and the fourth was composed by the *M. foetida* samples. Again, *R. moricandioides* appeared as the sister genus of *Moricandia* and *M. foleyi* accessions were grouped together with *Eruca* accessions. The *E. pinnatifida* accession AY722458 showed a distinctively long branch, probably indicating sequencing errors.

### Chronogram

There were no significant differences between the trees obtained with or without assuming a clock ($P = 0.92$; LnL $= -7851.663$ enforcing a clock; LnL $= -7835.925$ without assuming a clock). Therefore, the null hypothesis of equal evolutionary rates (i.e., molecular clock) was not rejected and these sequences can be assumed to follow a global molecular clock. With this assumption, we inferred chronograms using a constant rate of $1.2 \times 10^9$ substitution/site/year for the non-codifying plastidial DNA. The dating of the monophyletic Iberian clade was compatible with an origin during the end of the Messinian period (5.99–5.33 Ma). Other trees obtained with substitution rates reported in the literature (1.2–1.7 $\times 10^9$ s/s/y) were also compatible with this temporal window. Following, we produced a new chronogram constraining the origin of the Iberian clade to the Messinian salinity crisis period and including the ITS sequence of *M. sinaica* obtained from GenBank. This tree (Fig. 4) was congruent with that obtained with MrBayes (Fig. 1). The only topological differences were that the two samples of *M. foleyi* appeared to form a clearer lineage within the genus *Eruca*, and the position of *M. sinaica*, which was not included in the 24-samples set. In this chronogram the genus *Moricandia* separated 6.81 M years ago [5.83–7.89 Ma] from *Rytidocarpus.*

### DISCUSSION

The phylogenetic analyses presented here represent the evolution of the *Moricandia* genus clearly. Not only the Bayesian inference tree was concordant with the ML and timing trees, but also the cpDNA and ITS trees showed a fair agreement, with only minor discrepancies. In addition, all our samples were congruent with the GenBank accessions that were included in the ITS inclusive tree, although the latter tree showed lower resolution. In the trees of highest confidence, *Moricandia* appeared to be formed by two main lineages: the Iberian clade containing the Iberian species *M. moricandioides*, *M. foetida* and the new *M. rytidocarpoides;* and the African clade containing the species inhabiting the Southern Mediterranean region (*M. sinaica*, *M. suffruticosa*, *M. nitens*, *M. spinosa* and *M. arvensis*). The Bayesian inference trees showed high branch supports for this branching pattern and a good agreement with a molecular clock-like evolution. The timing of 7.19 Ma [5.18–9.23] for the separation of *B. rapa and R. sativus*, the two outgroup taxa in the chronogram, is

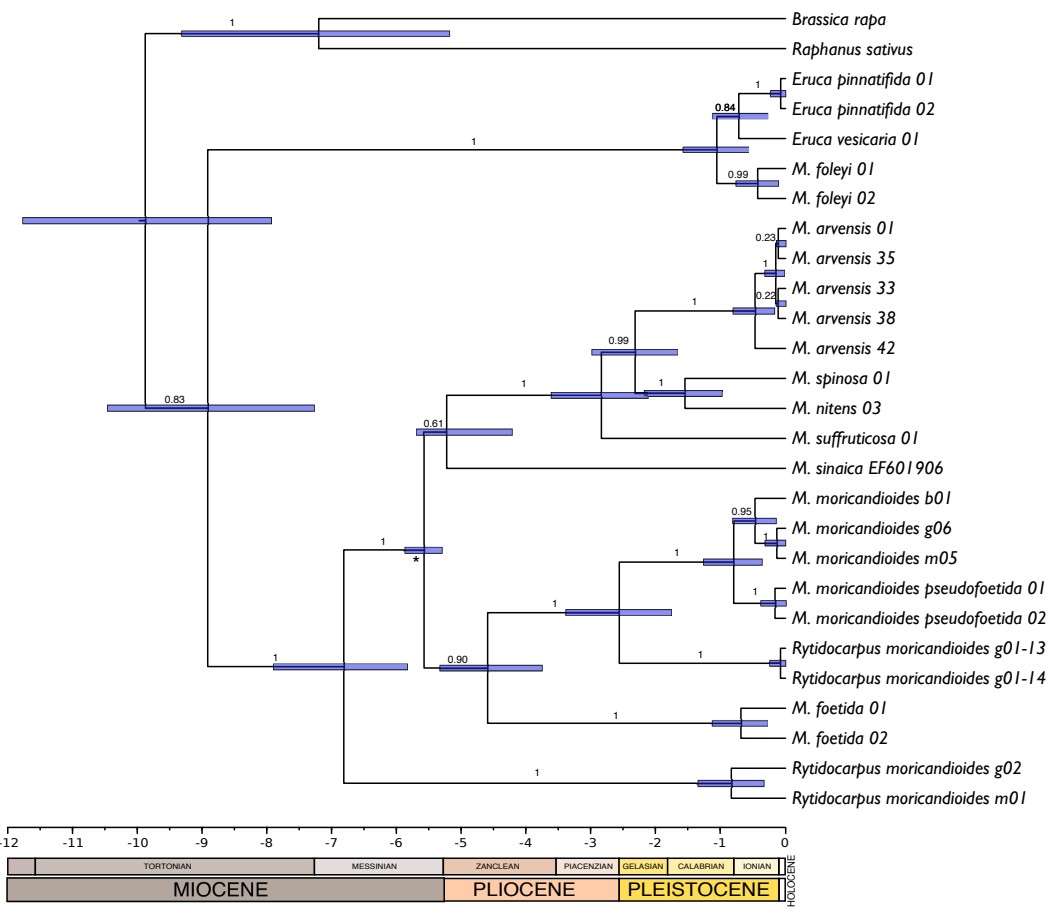

**Figure 4** **Chronogram obtained by Bayesian inference with the complete 24-samples set plus *M. sinaica*.** Branch labels refer to posterior probabilities. An asterisk marks the internal node used for calibration. Bars represent 95% confidence intervals. Temporal scale in Ma (million years ago).

congruent with the data previously reported (∼5.17 ± 2.5 Ma; *Hohmann et al., 2015*). This congruence is noteworthy since we used a strict clock that imposed a similar evolutionary rate in all lineages, which might not be realistic for the more distant (outgroup) taxa (*Drummond et al., 2006*). Therefore, we think that these phylogenetic trees are congruent and robustly represent the evolution of these taxa.

The genus *Moricandia* probably has a North African origin. Seven out of the previously eight recognized *Moricandia* species form a monophyletic group originated 6.81 Ma [5.83–7.89 Ma] after the separation from the *Rytidocarpus* lineage, their sister genera. Since *Rytidocarpus* is a North African endemism (*Maire, 1967*) and the genus *Moricandia* show a high number of species inhabiting North Africa (*Maire, 1967*), we support that *Moricandia* originated in North Africa, as was previously suggested (*Sobrino Vesperinas, 1978*). The colonization of the Iberian Peninsula probably occurred after a range expansion during the Messinian period, between 7.25 and 5.33 Ma (Fig. 5), coinciding with extensive African-Iberian floral and faunal interchanges (e.g., *Fernández-Mazuecos & Vargas, 2011*;

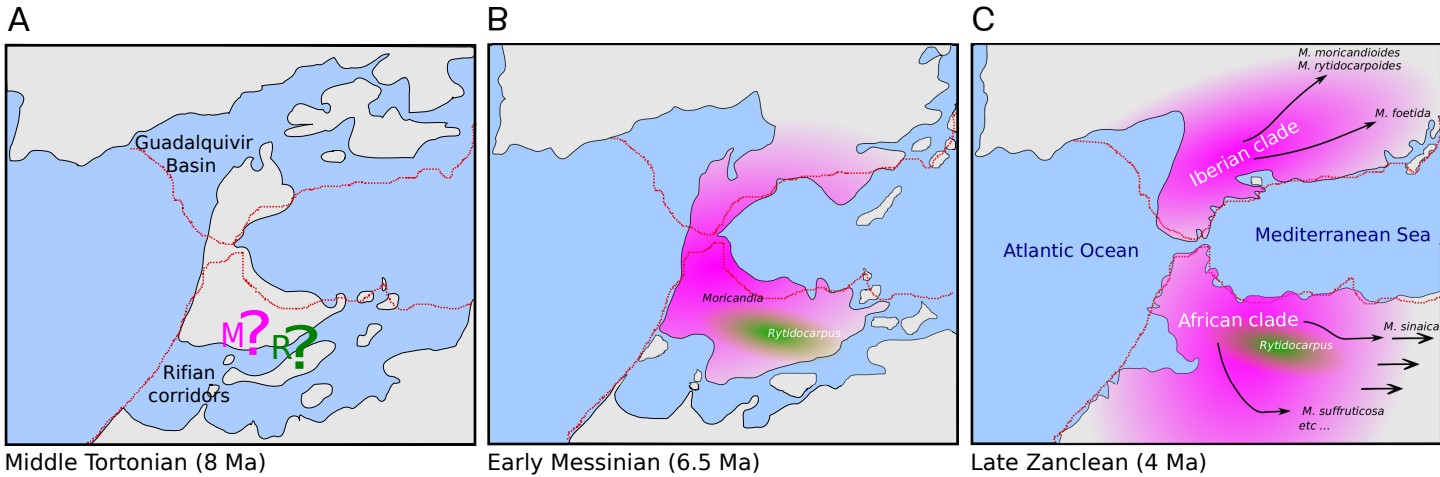

**Figure 5** Hypothesis of the biogeography of the genus *Moricandia* coupled to the geological events at the end of the Miocene (A: Middle Tortonian; B: Early Messinian) and early Pliocene (C: Late Zanclean) in the Betic-Rifean Arch, the Strait of Gibraltar at current times. Red lines depict the coastal lines at the present time. Based on the paleogeographical reconstruction of *Martín et al. (2009)*.

*Gibert et al., 2013*). In that period, land bridges between North Africa and South Iberia appeared due to tectonic uplift coinciding with the isolation of the Mediterranean Sea from the Atlantic Ocean circa 6.3 Ma (*Martín, Braga & Betzler, 2001*). After the new aperture of the Strait of Gibraltar during the Zanclean period, the Iberian and African clades were separated and began to diverge (Figs. 4 and 5). A similar biogeographical pattern has been found for other plants. For instance, the genus *Antirrhinum* produced several lineages separated by the Straight of Gibraltar (*Vargas et al., 2009*), as well as the genus *Hedera* and the *Saxifraga globulifera–reuteriana* complex (*Vargas et al., 1999*; *Vargas, Morton & Jury, 1999*). However, this is not a general biogeographical pattern since other species have maintained genetic connections along the two sides of the Straight of Gibraltar (*Rodríguez-Sánchez et al., 2008*).

*Moricandia sinaica* is the most basal taxon of the African clade, arising circa 5.22 Ma (Fig. 4), although, given the moderate support of this branching event, its phylogenetic position should be accepted with cautiousness. In the inclusive ITS tree, which included the GenBank-ITS set plus the 24-samples set, *M. sinaica* (inhabiting Egypt and West Asia) and *M. arvensis* var. *garamantum* (from South Algeria) appeared as close relatives. This fact suggests that these two taxa could be the same species, as it has been previously postulated (*Sobrino Vesperinas, 1984*). The rest of the species forming the African clade (*M. suffruticosa*, *M. nitens*, *M. spinosa* and *M. arvensis*) can easily hybridize (*Sobrino Vesperinas, 1997*) and have been included in the same cytodeme (*Prakash et al., 2009*). *Sobrino Vesperinas (1984)* postulated that the now widely distributed *M. arvensis* is the same as the *M. arvensis* var. *robusta* from the Constantine area in Algeria (*Gómez-Campo, 1999*). Unfortunately, the ITS sequences included here did not show enough resolution to support or deny this claim. In the inclusive ITS tree (Fig. 3), *M. arvensis* var. *robusta* appeared in the same clade as the other *M. arvensis* samples, but also forming a polytomy with *M. suffruticosa and*

*M. spinosa*. These two latter species show caryotypes with higher ploidy than the typical $2n = 28$ (*Sobrino Vesperinas, 1984*) of this genus (namely, *M. suffruticosa* with $2n = 56$ and *M. spinosa* with $2n = 84$; *Harberd, 1976*), but in the same ploidy series than the other *Moricandia* species.

The caryotype of *M. spinosa* suggests that this species could be a stabilized hybrid. We have found that *M. spinosa* shows a cpDNA closely related to those of *M. nitens* but for the nuclear ITS markers shows a more distant genetic relationship (Fig. 2). For these last markers, *M. spinosa* is more similar to *M. suffruticosa* (Fig. 2). These facts suggest that *M. spinosa* ($2n = 84$) is an amphidiploid species produced by whole genome duplication after a hybridization event between *M. suffruticosa* ($2n = 56$) and *M. nitens* ($2n = 28$). These kind of events have been pervasive in the evolution of this tribe and family (*Marhold & Lihová, 2006*; *Franzke et al., 2011*). However, more genetic analyses are necessary to confirm this hypothesis.

After the Zanclean aperture of the Strait of Gibraltar, the Iberian *Moricandia* clade diverged into three well-supported lineages that can be now identified as three genuine species (Figs. 4 and 5). Of these species, the endemic *M. foetida* was the first to diverge and is now completely reproductively isolated from *M. moricandioides*. In fact, *Sobrino Vesperinas* (*1993*, *1997*) experimentally demonstrated that they could not hybridize. The second lineage includes all the *M. moricandioides* samples in a monophyletic cluster. Five subspecies have been identified for this species based on morphological characters (*Sobrino Vesperinas, 1993*; *Sánchez Gómez et al., 2001*), with subspecies *pseudofoetida* showing intermediate characteristics with *M. foetida* (*Sánchez Gómez et al., 2001*). *Jiménez & Sánchez Gómez (2012)* based on ISSR markers proposed that *M. moricandia pseudofoetida* arose by reproductive isolation rather than hybridization between *M. foetida* and *M. moricandioides moricandioides*. Our current results do not refute that hypothesis and show that this subspecies, that inhabit similar environments that *M. foetida*, is clearly a *M. moricandioides* subspecies without showing signal of past hybridization.

The third species in the Iberian clade are plants growing in marls and calciferous substrates in a few places of the Guadiana Menor River Basin. Based mostly on fruit morphology, these plants were originally ascribed to the Moroccan-endemism *Rytidocarpus moricandioides* by *Morillas-Sánchez & Fernández López (1995)*, and later accepted in the Vascular Flora of Eastern Andalusia (*Blanca et al., 2011*). These plants differ from other *Moricandia* in some peculiar morphological characters. Namely, they present sepals with scarious margins that persist during fruit development and fruits with two segments, the upper with beak shape, being similar to *Rytidocarpus*, whereas the other *Moricandia* species present non-persistent sepals without scarious margins and fruit in a siliqua. However, despite these morphological differences, our phylogenetic study clearly shows that they belong to the genus *Moricandia* (Figs. 1–4), inside the Iberian clade as a sister species to *M. moricandioides*. In addition, an exhaustive morphological comparison between the Jaén and the Morocco specimens clearly separated the Spanish samples from the genuine, Moroccan, *Rytidocarpus moricandioides* (see Tables S2, S3 and Figs. S3 S4). Therefore, these plants should be ascribed to the genus *Moricandia* and they deserve the taxonomic rank of species due to both their phylogenetic position and their distinctive morphological

traits. We proposed the name *Moricandia rytidocarpoides* to denominate this new species *Moricandia rytidocarpoides* Lorite, Perfectti, Gómez, González-Megías & Abdelaziz sp.nov. urn:lsid:ipni.org:names:77166015-1 (see Appendix for a formal taxonomical description). Consequently, the siliqua fruits and the dehiscent scarious sepals are no longer defining characteristics (i.e., diagnostic traits) of the genus *Moricandia*. Several molecular analyses have demonstrated that fruit morphology shows homoplasy in Brassicaceae (*Appel & Al-Shehbaz, 2003*; *Koch, Al-Shehbaz & Mummenhoff, 2003*), and, therefore, fruit traits are not good indicators of phylogenetic relationships despite they have been widely used as taxonomic diagnostic traits (*Hall et al., 2006*).

The taxonomic status of *Moricandia foleyi* should be also amended. In all the phylogenetic analyses performed here *M. foleyi* appeared outside the *Moricandia* lineage and within the lineage of the *Eruca* species. *Eruca* is a genus of annual, non-hetero-arthrocarpic plants with a controversial taxonomy. Depending on the taxonomical treatment, *Eruca* includes from one to four species. *Gómez-Campo (1999)* considered this genus as monotypic, after *E. loncholoma* was ascribed to *Brassica* subgen. *Brassicaria* (*Gómez-Campo, 1999*) and *E. setulosa* was moved to the proposed genus *Guenthera* (*Gómez-Campo, 2003*). *Pratap & Gupta (2009)* also supported this view, meanwhile *Warwick, Francis & Gugel (2009)* recognized four taxa: *E. loncholoma*, *E. pinnatifida*, *E. setulosa* and *E. vesicaria*. However, after *E. pinnatifida* was classified as a subspecies of *E. vesicaria*, http://www.theplantlist.org/ accepted only three species in this genus: *E. loncholoma*, *E. setulosa* and *E. vesicaria*, with this last species presenting three subspecies. Of these subspecies, two (*E. vesicaria vesicaria* and *E. vesicaria pinnatifida*) are circumscribed to the West Mediterranean region, whereas the subspecies *sativa* shows a circum-mediterranean distribution (*Pignone & Gómez-Campo, 2011*), although it is currently cultivated in many other areas of the world (*Gómez-Campo & Prakash, 1999*). Our phylogenetic analyses support the inclusion of *M. foleyi* in the genus *Eruca* (Figs. 1–4). In addition, other evidences separate *M. foleyi* from the genus *Moricandia*. This species has been reported to be $2n = 14$ (*Sobrino Vesperinas, 1997*), clearly different from the caryotipic values reported for the rest of the *Moricandia* species ($2n = 28, 56, 84$; *Sobrino Vesperinas, 1984*), but also a different value when compared to the *Eruca* species ($2n = 22$; *Harberd, 1976*). Recently, in an ITS phylogeny, *Schlüter et al. (2016)* separated *M. foleyi* of the *Moricandia* lineage, although unfortunately they did not include in their analyses any *Eruca* samples. Anecdotally, *Maire (1967)* described *M. foleyi* as a glabrous and robust green herb with a strong smell of *E. vesicaria* (''à forte odeur d'*Eruca vesicaria*''). In our nuclear-sequences trees (24-samples and the GenBank-ITS sets) *M. foleyi* consistently appeared in the *Eruca* clade. The same pattern appeared in the cpDNA tree (see Fig. 2 and Fig. S2), indicating that both nuclear and cytoplasmic genes ascribe *M. foleyi* to the genus *Eruca* and excluding a recent hybridization event. Therefore, *M. foleyi* should be removed from the genus *Moricandia* and ascribed, at least provisionally, to the genus *Eruca*: *Eruca foleyi* (new combination): *Eruca foleyi* (Batt.) Lorite, Perfectti, Gómez, González-Megías & Abdelaziz comb. nov. urn:lsid:ipni.org:names:77166016-1 (see Appendix for a formal taxonomical description).

## CONCLUSIONS

We have reported an inclusive dated phylogeny of the genus *Moricandia,* showing that, after the adscription of *M. foleyi* to the genus *Eruca*, it is a recent monophyletic genus that evolved in North Africa between 5.81 and 7.89 Ma diverging from its sister monotypic genus *Rytidocarpus.* Following the new aperture of the Strait of Gibraltar during the Zanclean period, *Moricandia* diverged in two different lineages: the Iberian and Afrian clades. At a finer scale, future work should address the evolutionary relationships between the different subspecies of *M. arvensis* and confirm the phylogenetic position of *M. sinaica*.

## ACKNOWLEDGEMENTS

We are grateful to Cristina Sánchez-Prieto and Raquel Sánchez for their help in DNA extraction and amplification, Isabel Cañabate for facilitating sampling in Olula del Río and Isabel Reche for the critical reading of the MS. FP thanks CJ Rothfels for his support during a sabbatical stay at UC Berkeley.

## APPENDIX. PROPOSED TAXONOMIC CHANGES

### *Moricandia rytidocarpoides*

The results gathered from phylogenetic data and morphological traits (both quantitative and qualitative, see Tables S2, S3 and Fig. S3, S4) led us to conclude that southeastern Spanish populations of (the previously ascribed) *R. moricandioides* constitute a new species, clearly separately from *R. moricandioides* Coss.

*Moricandia rytidocarpoides* Lorite, Perfectti, Gómez, González-Megías & Abdelaziz *sp. nov.*

*Holotype:* Spain: Jaén, Guadiana Menor basin Quesada, close to El Salón, 473 m a.s.l. 37°48′19.51″N/3°09′01.31″W, marly slopes over bad lands, 04/04/2017, Leg. and Det. J. Lorite. GDA62636.

*Diagnosis:* It differs from Rytidocarpus moricandioides Cosson in many morphological characters. Among them, we remark the shorter (9.5–16 mm length) and markedly reticulated fruits, shorter flowers (7–23 mm) and shorter low leaves (13–175 × 2–81 mm) with lobate to pinnatifid margin.

*Description:* Annual plant of 30 (10–75) cm, glabrous, usually with erect 1–2 flowered stems not ramified. Lower leaves of 13–175 × 2–81 mm, rosette forming, sub-sessile with lobed to pinnatifid margin. Upper leaves 10–14 × 0.9–12 mm, chordate to sagittate, amplexicaul, markedly acute at the apex. Inflorescence racemose, no flexuose, with c. 60 (30–80) flowers. Flowers arranged in simple and terminal racemes with 4–14 flowers, actinomorphic, hermaphroditic, and tetrameric. Sepals of 7.1–10.3 × 1–2.5 mm, green-light purplish, acute at the apex and gibbous at the base. Petals of 13 (7–23) mm, long clawed, light purplish. Six tetradynamous stamens with long filaments of 7–14 mm and short ones of 5–12 mm. Fruits in silique of 9.5–16 × 3–3.4 mm with stigma bilobed, with 19 (11–30) seeds of 1.1–1.65 × 0.4–1.1 mm. Fruit valve markedly reticulate with one central nerve, with beak of 6–8 mm.

*Flowering time:* March–May.

*Fruiting time:* March–May.

*Habitat description and distribution:* Inhabits dry and semiarid badlands over gypsiferous marls, in gaps of an open semiarid scrublands composed of *Lygeum spartum* L., *Hammada articulata* (Moq.) O. Bolós & Vigo, *Salsola vermiculata* L., *Suaeda vera* J.F. Gmelin, *Macrochloa tenacissima* (L.) Kunth, *Thymus vulgaris* L., *Frankenia thymifolia* Desf., *Asparragus horridus* L., and *Urginea maritima* (L.) Baker. The species appears in small patches along the Guadiana Menor River, between 400 and 650 m asl. It is a rare species due to the narrow distribution area, the patchy habitat that occupies, and because it shows the typical extreme population fluctuations of the annual plants in such semiarid environments. According the IUCN categories the species has Data Deficient (DD), and, therefore, it is crucial to evaluate the species at short term.

Ethymology: The noun *rytidocarpoides* refers to the characteristic fruit valve resembling those of *Rytidocarpus moricandioides*.

## *Eruca foleyi*

*Eruca foleyi* (Batt.) Lorite, Perfectti, Gómez, González-Megías & Abdelaziz *comb. nov.* (*Basionym* = *Moricandia foleyi* Batt. Bull. Soc. Bot. France 61: 52. 1914)

*Holotype:* Algeria: Sud oranais, Nebkas dans les vallées de l'Oued Namous, et de la Zousfana, Haute Saoura, Bejjig. 04/1913, Leg: H. Foley, MPU006516.

*Diagnosis:* It differs from *Moricandia* species in many morphological characters. Among them, we remark that this species is sparsely hairy, with hairy fruits similar in shape to the genus *Eruca*. Molecular data place this taxon within genus *Eruca*. As a result, we propose that this taxon must be transferred from *Moricandia* to *Eruca* genus.

*Description:* Annual plant up to 150 cm high, sparsely hairy, usually with erect 1–5 flowered stems not ramified. Lower leaves of 50–250 × 15–50 mm, rosette forming, sessile, and lobulated. Upper leaves oblong attenuate, no amplexicaul with crenate margin. Inflorescence corymbose on anthesis, no flexuose, with c. 20–30 (up to 200) flowers. Flowers actinomorphic, hermaphroditic, and tetrameric. Sepals of 7–9 mm, green with scarious margin. Petals of 20 (15–22) mm, long clawed, purplish. Six tetradynamous stamens with long filaments of 11–13 mm and short ones of 6–10 mm. Fruits in silique of 30–55 × 2,5–3(3) mm, with a pedicel of 6–11 mm, with hairy valve. Seeds 30–40, biseriate, of 1.6–1.7 × 0.8–0.9 mm.

*Flowering time:* January–March.

*Fruiting time:* March–April.

*Habitat description and distribution:* Inhabits desert plains, valleys; dry stream beds, fields and crops; muddy, sandy alluvium, sandstone and shale in Northern Sahara (Algeria and Morocco).

### Funding

This study was supported by grants from the Spanish Ministerio de Economía y Competitividad (CGL2013-47558-P), including EU-FEDER funds, and Plan Andaluz de Investigación (P11-RNM-7676 and P08-RNM-0381). Mohamed Abdelaziz has been supported by TransSpeciation project (CGL2014-59886-JIN) from the Spanish Ministerio de Economía y Competitividad, including EU-FEDER funds. There was no additional external funding received for this study. The funders had no role in study design, data collection and analysis, decision to publish, or preparation of the manuscript.

### Grant Disclosures

The following grant information was disclosed by the authors:
Spanish Ministerio de Economía y Competitividad: CGL2013-47558-P, CGL2014-59886-JIN.
EU-FEDER funds.
Plan Andaluz de Investigación: P11-RNM-7676, P08-RNM-0381.

### Competing Interests

The authors declare there are no competing interests.

### Author Contributions

- Francisco Perfectti conceived and designed the experiments, performed the experiments, analyzed the data, contributed reagents/materials/analysis tools, wrote the paper, prepared figures and/or tables, reviewed drafts of the paper.
- José M. Gómez conceived and designed the experiments, performed the experiments, analyzed the data, contributed reagents/materials/analysis tools, wrote the paper, reviewed drafts of the paper.
- Adela González-Megías and Mohamed Abdelaziz performed the experiments, contributed reagents/materials/analysis tools, reviewed drafts of the paper.
- Juan Lorite performed the experiments, analyzed the data, contributed reagents/materials/analysis tools, wrote the paper, prepared figures and/or tables, reviewed drafts of the paper.

### DNA Deposition

The following information was supplied regarding the deposition of DNA sequences:

The ITS sequences have been submitted to GenBank with accession numbers MF192766–MF192789.

The *ndhF* sequences have been submitted to GenBank with accession number: MF192790–MF192813.

The *trnT-trnF* sequences have been submitted to GenBank with accession number: MF192814–MF192837.

Alignments used in the phylogenetic analyses are available as Supplementary Files.

## Data Availability

The alignment of the sequences used in this manuscript are available as Supplemental Files.

## New Species Registration

The following information was supplied regarding the registration of a newly described species:

*Moricandia rytidocarpoides* (new species):

*Moricandia rytidocarpoides* Lorite, Perfectti, Gómez, González-Megías & Abdelaziz sp.nov. 77166015-1

*Eruca foleyi* (new combination):

*Eruca foleyi* (Batt.) Lorite, Perfectti, Gómez, González-Megías & Abdelaziz comb. nov. 77166016-1.

## Supplemental Information

Supplemental information for this article can be found online at http://dx.doi.org/10.7717/peerj.3964#supplemental-information.

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
