# Peer review of "Molecular phylogeny and evolutionary history of Moricandia DC (Brassicaceae)"

_PeerJ, doi:10.7717/peerj.3964_

## Round 0.1 · original submission · Major Revisions

The authors present interesting research that contributes to the evolution and systematics of the genus Moricandia. However I must ask the authors to please follow the suggestions of the reviewers carefully to improve the solidity of the new knowledge presented to improve the science of systematic and evolution of Brassicaceas, before manuscript definitive acceptance.

I attach the comments of the reviewers, The English should be improved substantially and the authors need to improve the support value of Bayesian tree and ML tree. In particularly the authors have to follow the improvements requested by Reviewer Number two who suggested that the authors do the AU-test of other potential topologists of Moricandia species using the same dataset as in Figure 1.

Also some technical details - technical terms including bp (base pair) are used differently as "nt" or terms its1, its2 (page 8 line 140) instead of ITS1, ITS2, as is mention by reviewers number 3.

I think the reviewers and authors has done a nice job with the article contribution which will be a lot better after improvements.

Then please get to work ("manos a la obra"). I hope to hear from you soon.

Be in touch.
Hugo

Reviewer 1 ·

Basic reporting

This manuscript explores the phylogenetic relationships of Moricandia species (seven spp.) Sampling seems to be adequate with several specimens per species and nuclear ITS and plastid trnT-F and ndhF sequences. Author also tested the position of the closely related Rytidocarpus moricandioides. Overall, analyses are adequate, and results obtained were interesting, revealing that M. foleyi is not included in Moricandia. Moricandia s.s. (without M.foleyi) is composed by two geographically structured clades, the African and the Iberian clades. Additionally, Iberian specimens of R. moricandioides are included in the Iberian clade, while African specimens (Rytidocarpus s.s) were placed sister to Moricandia. Result from divergence time analyses related divergence of Moricandia with the geologic history of the region; however I am not an expert on the geological evolution of this area and cannot give a robust evaluation of these findings.

I think that this manuscript is adequate for its publication in PeerJ after applying minor changes (see comments for the author)

Experimental design

no comment

Validity of the findings

no comment

Additional comments

This manuscript explores the phylogenetic relationships of Moricandia species (seven spp.) Sampling seems to be adequate with several specimens per species and nuclear ITS and plastid trnT-F and ndhF sequences. Author also tested the position of the closely related Rytidocarpus moricandioides. Overall, analyses are adequate, and results obtained were interesting, revealing that M. foleyi is not included in Moricandia. Moricandia s.s. (without M.foleyi) is composed by two geographically structured clades, the African and the Iberian clades. Additionally, Iberian specimens of R. moricandioides are included in the Iberian clade, while African specimens (Rytidocarpus s.s) were placed sister to Moricandia. Result from divergence time analyses related divergence of Moricandia with the geologic history of the region; however I am not an expert on the geological evolution of this area and cannot give a robust evaluation of these findings.

I think that this manuscript is adequate for its publication in PeerJ after applying minor changes (see comments for the author)

some suggestions/ corrections:

Maybe it will be better if results from individual analyses of ITS and cpDNA are showed before the combined analyses

Authors could indicate on Fig. 4 (the chronogram) the node used for the calibration based on the geological information (uniform 5.39-5.33 mya)

I am not an expert in the tribe Brassiceae, but the authors transferred M. foleyi to the genus Eruca because this species resulted sister of Eruca. However phylogenetic analyses included only four Eruca species, Brassica rapa, and Raphanus sativus. Is this sampling adecuate? Perhaps, in analyses using a broader sampling of Brassiceae M. foleyi could be recovered related to other genus of the tribe.

It would be interesting to analyze this dataset using species tree inference under the multispecies coalescent model

Line 182: GTR+G instead GMT+G

Reviewer 2 ·

Basic reporting

Francisco Perfectti et al. reported a research to reconstruct the phylogenetic relationships of Moricandia species using the traditionally marker genes of ITS and trnT-trnF. I think this paper is ok and I only have several following concerns.

1. To better present the results, the genus name and clades name of 'African clade' and 'Iberian calde' should be labeled on the Figure 1.

2. I can not understand 'The subspecies M. moricandioides pseudofoetida appeared inside the branch of M. moricandioides but well separated from the other M. moricandioides subspecies' on page 13 line 240-241. Actually, two M. moricandioides pseudofoetida species are sister group of other three M. moricandioides subspecies.

3. on page 13 line 245, the author should specify the exact samples name of the R. moricandioides.

4. The Fig. 4 should be Fig. 3 on page 14 line 271.

Experimental design

5. Normally, the support value of Bayesian tree pretend to be higher than ML tree. The author should not only label the support value of Bayesian tree, but also label the support value of ML tree on Figure 1 by only presenting the topology obtained by the Bayesian method.

6. The authors only listed the Bayesian trees of separated results on Figure 2. How about the ML tree of separated cpDNA and ITS?

Validity of the findings

7. The authors claimed that two Rytidocarpus moricandioides subspecies are located into the clade of Moricandia and is indeed a new Moricandia species. But how about the statistical possibilities of other topologies? I suggested the authors to do the AU-test of other potential topologists of Moricandia species using the same dataset as in Figure 1. Unless the result can reject the possibility that the Rytidocarpus moricandioides g01-13 and g01-14 are sister groups of all other species in the pink filled region, I can trust that these two species are actually new Moricandia species.

Additional comments

no comments

Reviewer 3 ·

Basic reporting

The paper provides some updates on the evolution and systematics of the genus Moricandia. Results and conclusions look reliable and the paper is in focus of
the journal. On the other hand, the quality of the text is poor and needs substantial English edition (eg. p 6 line 98. Please replace "their" by "its"). I strongly suggest that you have a native English speaking colleague review your manuscript. Additionally some technical terms including bp (base pair) used differently as "nt" or autor use the terms its1, its2 (page 8 line 140) instad of ITS1, ITS2 (these are commen usage of these regions and also the name of the relevant primers).

Materials used in this study look proper for such kind of phylogenetic studies as well as literature references.

Experimental design

Design of study looks good and proper for filling the gaps regarding phylogeny and systematics of the Moricandia with the exception one analyses. For details, please see comment in "Validity of findings " section.

Regarding model selection, i have never heard the GTM model that autors used in Bayesian analyses. Please check it.

Validity of the findings

Data looks robust but i do not understand why Ancestral Area Recounstruction analyses did not include this comprehensive study. Without these analyses all discussion about origin of the Moricandia species will be speculative.

---

## Round 0.2 · accepted · Accept

Thank you for the effort to improve the article

Take care

Hugo